computer modelling and simulation/materials science/nanotechnology

Al–4 at.%Cu alloy, homogeneous nucleation, isothermal solidification, molecular dynamics

**Author for correspondence:**
Xiangge Qin
e-mail: qinxiangge@jmsu.edu.cn

# Research on homogeneous nucleation and microstructure evolution of aluminium alloy melt

## Lan Zhan, Mingzhong Wu and Xiangge Qin

School of Materials Science and Engineering, Jiamusi University, 258th Xuefu Street, Xiangyang District, HeiLongJiang 154007, People's Republic of China

LZ, 0000-0001-7223-2124

In this paper, based on the embedded atom method (EAM) potential, molecular dynamics simulations of the solidification process of Al–4 at.%Cu alloy is carried out. The Al–Cu alloy melt is placed at different temperatures for isothermal solidification, and each stage of the entire solidification process is tracked, including homogeneous nucleation, nucleus growth, grain coarsening and microstructure evolution. In the nucleation stage, the transition from high temperature to low temperature manifests a change from spontaneous nucleation mode to divergent nucleation mode. The critical nucleation temperature of the Al–Cu alloy is determined to be about 0.42 $T_m$ ($T_m$ is the melting point of Al–4 at.%Cu) by calculating the nucleation rate and the crystal nucleus density. In the nucleus growth stage, two ways of growing up are observed, that is, a large crystal nucleus will absorb a smaller heterogeneous crystal nucleus, and two very close crystal nuclei will merge. In the microstructure evolution of the isothermally solidified Al–Cu alloy, it is emerged that the interior of all nanocrystalline grains are long-period stacking structure composed of face centred cubic (FCC) and hexagonal close-packed (HCP). These details provide important information for the production of Al–Cu binary alloy nano-polycrystalline products.

## 1. Introduction

2XXX series aluminium alloy (Cu element content is 3.8–4.9 at.% [1]) as a very important alloy material, has long served in shipbuilding industry [2], automobile industry [3,4], aerospace [5,6] and other fields. Especially in recent years, the rise of laser cladding and three-dimensional printing manufacturing processes has promoted Al–Cu alloys to become irreplaceable metal materials. It is known that these processing procedures are inseparable from

the solidification process. However, it is still difficult to accurately control the microstructure of metals and alloys during the solidification process. Because solidification is affected by many aspects of physics, such as heat transfer, convection and solute diffusion. In addition, if the crystal nucleus can be formed under a large degree of supercooling, it can also produce ultra-fine or even nanocrystalline structures [7]. Therefore, it is important to control and improve the solidification process to ensure and improve the quality of metals and alloys products.

So far, experimental studies [8–11] on the nucleation and growth behaviour of aluminium alloys have mainly adopted optical microscopy (OM), scanning morphology characterization, electron microscopy (SEM) and *in situ* transmission electron microscopy (TEM) to achieve solidification structure analysis. For example, Zweiacker [12] used precession electron diffraction to assist automated crystal orientation mapping for TEM and scanning TEM (STEM) research. Al–4 at.%Cu alloy is rapidly solidified, and fine equiaxed crystals are observed. The latest research of Xu *et al.* [13] used *in situ* micro-focus X-radiography technology to observe the nucleation and growth dynamics of plate-like thin samples of hypereutectic Al–Si(–Cu) alloy in real time. However, it is still not easy to directly observe and analyse the crystal embryo formation and crystal nucleus growth during the isothermal homogeneous nucleation process of the melt using the existing experimental methods.

Molecular dynamics (MD), as a classical particle method that can calculate the evolution of atom/molecule configuration, has been exploited to study the solidification of systems ranging from thousands of atoms to billions of atoms. Relevant research [14–19] in recent years evidence that there is good agreement between MD prediction data and crucial nucleus size, nucleation temperature experiments and classical nucleation theory (CNT) results. Hou *et al.* [20] first simulated the solidification process of bulk Al and discussed the effect of cooling rate on the solidification microstructure of pure aluminium. It was observed that the face centred cubic (FCC) and hexagonal close-packed (HCP) structures coexist in the crystal structure. Subsequently, the aluminium melt was still the focus of research. Since isothermal solidification can obtain the structure formation process of crystal nucleation rate, critical crystal nucleus size and grain coarsening, the process of isothermal solidification is often adopted in experiments to study the solidification structure of materials [21,22]. Therefore, the researchers adopted different atomic potentials to carry out further research on isothermal solidification and homogeneous nucleation, and obtained important solidification data of bulk pure aluminium [23,24]. With the rapid development of computing power, Shibuta *et al.* [25] performed the largest MD simulation in the field of solidification process (1 billion atoms), and confirmed that there are both homogeneous nucleation and local heterogeneous nucleation during the isothermal solidification of Fe melt. Furthermore, the liquid–solid transition of alloy systems has also received attention. Lin *et al.* [26] studied the heterogeneous nucleation of Al–Cu liquid alloys with different copper content on the copper matrix, and proposed that the slot angle of the matrix would also affect the heterogeneous nucleation of the structure. Mahata *et al.* [27], as the first researcher using MD to understanding the directional solidification of aluminium alloys, explored Al–11 at.%Cu alloy directional solidification nucleation and growth process and predict the mechanical properties of the directional solidification structure. It provides important information for the in-depth study of aluminium alloy directional solidification. In addition to studying the solidification behaviour of the block structure, the researchers also studied the solidification characteristics of quasi-two-dimensional samples. For example, Shibuta *et al.* [15] and Sui *et al.* [28], respectively, employed MD simulations to prepare quasi-two-dimensional iron and polysilicon samples, and studied the nucleation and microstructure evolution under isothermal solidification. Therefore, the study of the solidification law of Al–Cu binary alloy nano-polycrystalline will be of great significance.

In the current research, MD is employed to study the process of isothermal solidification of Al–4 at.% Cu alloy melt. By tracing the isothermal solidification process of the melt, the homogeneous nucleation, crystal nucleation growth, grain coarsening and microstructure were analysed. It is expected to predict the general law of the isothermal solidification process of Al–4 at.%Cu alloy plate-like thin samples.

## 2. Simulation methods

The MD simulation of Al–4 at.%Cu homogeneous nucleation are completed by the large atomic/molecular mass parallel simulator (LAMMPS) [29]. First, a simulation system of $32.4 \times 32.4 \times 0.8$ nm is established, with 51 200 atoms. The simulation system is equivalent to intercepting a quasi-two-dimensional sample at the centre of the actual large-scale solidified structure; 4 at.%Cu (about 2048 atoms) is randomly distributed in aluminium. Figure 1 indicates the initial size of the simulation

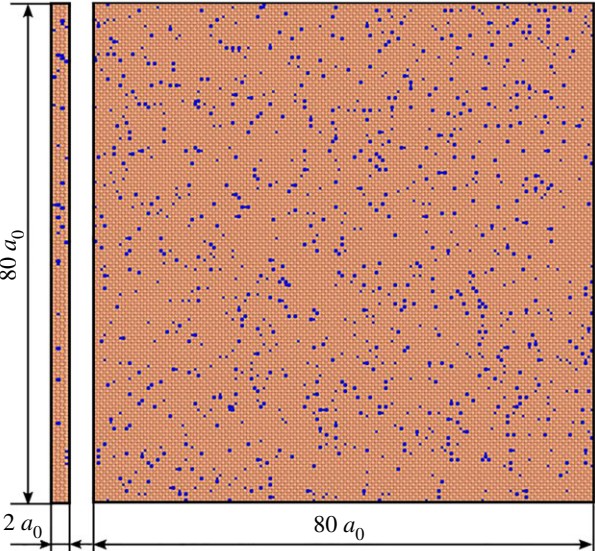

**Figure 1.** Molecular dynamics model of the homogeneous nucleation and growth process of Al–4 at.%Cu melt. Orange represents Al atoms, blue represents Cu atoms and $a_0$ represents the lattice constant of Al.

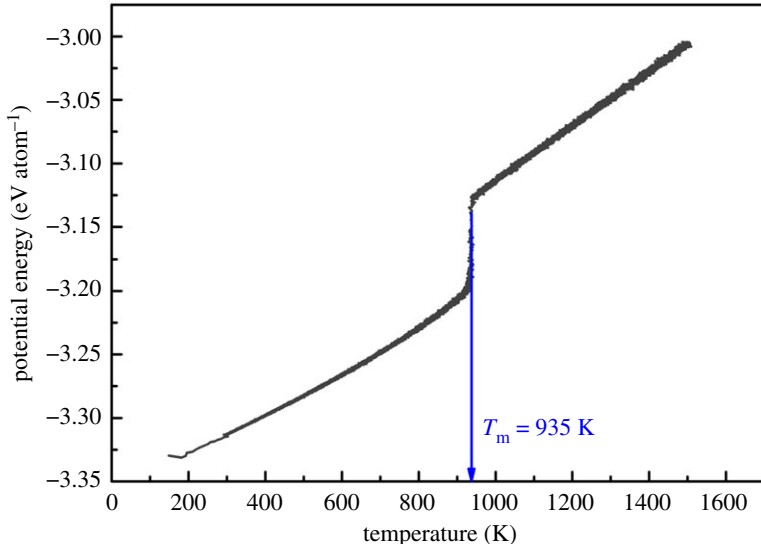

**Figure 2.** The potential energy per atom of Al–4 at.%Cu as a function of temperature. $T_m$ represents the melt point.

frame ($a_0$ is the lattice constant of Al). The interaction between aluminium and copper is described by the embedded atom method (EAM) potential, which has been widely applied in the interaction of metals and alloys. The AlCu EAM potential comes from Cai [30]. We first used the interaction potential to calculate the melting point according to the model used in this study, and obtained the curve of the average potential energy of the atoms changing with temperature during the heating process of the Al–4 at.% Cu alloy, as shown in figure 2. The curve in the figure suddenly rises at 935 K, breaking the linear increase, which means that a phase change has occurred here. This temperature is defined as the melting point of the system, and the error of the phase diagram data (923 K) is about 0.8%.

In order to obtain a completely melted Al–Cu alloy without FCC crystals, the FCC crystal Al–Cu simulation box is heated from 300 K to 1.48 $T_m$ ($T_m$ is the melting point of Al–4 at.%Cu) and relaxed for 200 ps. Then, the uniform Al–Cu liquid is isothermal at the target different temperatures (0.6 $T_m$, 0.54 $T_m$, 0.48 $T_m$, 0.42 $T_m$, 0.39 $T_m$). The system performs isothermal solidification at the target temperature, and the isothermal time is at least 1000 ps. Periodic boundary conditions are employed throughout the MD simulation process. The temperature and pressure are controlled by a Nose–Hoover thermostat and a Parrinello–Rahman barostat [31], and the system pressure is placed at zero.

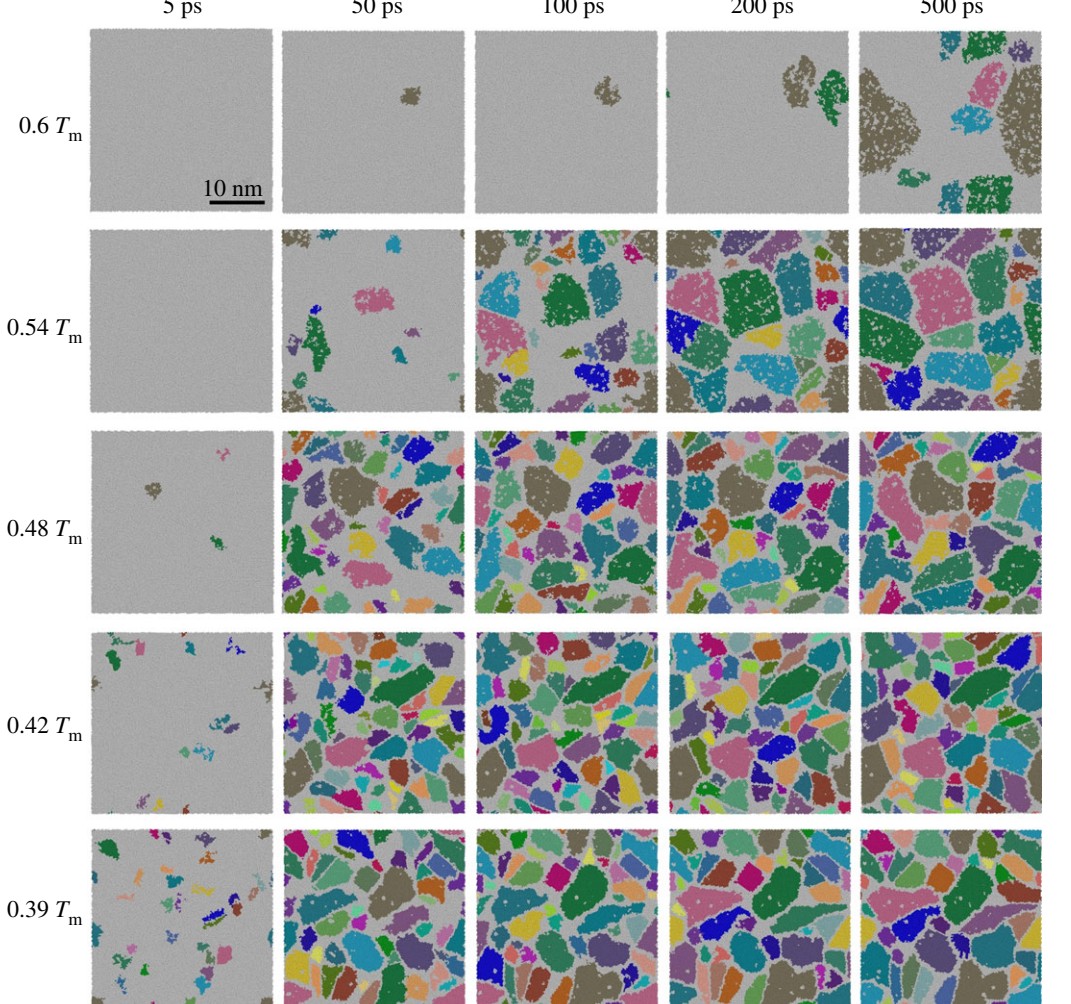

**Figure 3.** Snapshots of the homogeneous nucleation and growth process of Al–4 at.%Cu melt at different temperatures. The white atoms represent melt atoms or grain boundaries atoms. Each colour in the coloured atoms represents a crystal grain.

The time step is 1 fs, and data is collected every 1 ps. Each simulation is repeated five times to ensure the stability of the simulation data. The open visualization tool (OVITO) [32] is adopted to monitor the melting, nucleation and solidification process of the Al–Cu alloy. Grain segmentation as a new feature of OVITO is applicable to the statistics of the number of grains. This algorithm produces good segmentation in a range of microstructure types. It performs well on low-angle grain boundaries (GBs) and can discriminate between grains and sub-grains. Dislocation extraction algorithm (DXA) [33] analysis is used to check grain orientation.

## 3. Results and discussion

### 3.1. Isothermal solidification process of Al–4 at.%Cu alloy

Figure 3 displays the nucleation process of the Al–Cu alloy melt at the temperature of $0.6\,T_m$, $0.54\,T_m$, $0.48\,T_m$, $0.42\,T_m$ and $0.39\,T_m$ during equilibrium solidification for 500 ps. Grain segmentation is employed to segment and colour the grains, and each colour represents a grain. In figure 3, it can be observed that the incubation period required for the formation of crystal nuclei becomes shorter as the degree of supercooling increases, that is, the increase of the degree of supercooling provides the energy required for nucleation. In the case of $0.6\,T_m$, only one crystal nucleus is formed in continuous isothermal 50 ps, and the incubation period is the longest. As the isothermal time continues, when the nuclei grow, new nuclei are formed in sequence, reaching 500 ps to form 7 crystal nuclei. At $0.54\,T_m$, some crystal nuclei are generated before 50 ps. During the isothermal process, a few new crystal

royalsocietypublishing.org/journal/rsos　　R. Soc. Open Sci. **8**: 210501

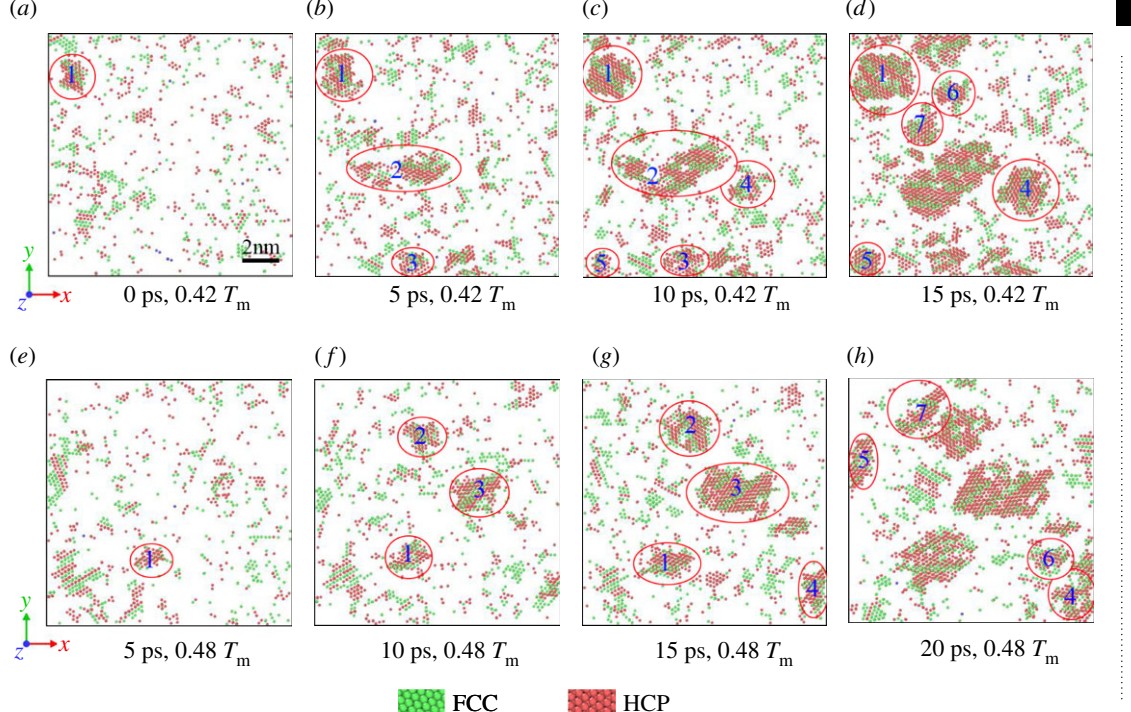

**Figure 4.** Snapshots of number of nuclei are shown at various time steps for 0.42 $T_m$ (388 K) and 0.48 $T_m$ (443 K) isothermal solidification. Green represents FCC, red represents HCP. (a) 0 ps, crystal nuclei have been formed, and then (b) 5 ps, (c) 10 ps and (d) 15 ps new crystal nuclei appear in sequence. At a higher isothermal temperature of 0.48 $T_m$, (e) 5 ps, the first crystal nucleus appears, and then every 5 ps shows nucleation; (f) 10 ps, (g) 15 ps and (h) 20 ps all show new crystal nuclei generate.

nuclei are formed near the previously generated crystal nuclei. The nucleation mode at 0.6 $T_m$ and 0.54 $T_m$ temperature is spontaneous nucleation. Finally, a nano-polycrystalline structure of Al–Cu alloy is formed. Next, the isothermal nucleation process of 0.48 $T_m$, 0.42 $T_m$ and 0.39 $T_m$ are inspected. As the temperature decreases, the degree of subcooling increases, the incubation period for crystal nucleation becomes shorter and the number of crystal nuclei increases. A large number of crystal nuclei are formed in a short time isothermally, and the growth of these nuclei is accompanied by the formation of a large number of new nuclei. This nucleation method corresponds to the low-temperature divergent nucleation [25,34]. The existing crystal nuclei are formed within 50 ps and grow rapidly. However, during the growth process at 0.39 $T_m$, although the number of primary crystal nuclei is large, part of crystal nuclei during the growth process are eliminated because it does not reach the critical crystal nucleus size. In the end, the number of crystal nuclei decreases, and the crystal grains are larger than the crystal grains at a temperature of 0.42 $T_m$. Therefore, the crystal grains under the isothermal solidification condition of 0.42 $T_m$ are the smallest.

## 3.2. Homogeneous nucleation of Al–4 at.%Cu melt

In figure 4, 0.42 $T_m$ and 0.48 $T_m$ are taken as examples to observe the homogeneous nucleation process. Polyhedral template matching (PTM) [35] is employed to colour the atoms, green represents FCC lattice, and red represents HCP lattice. As captured in figure 4, at different temperatures, certain nuclei will nucleate and grow at a specific time step, which is dominant at lower temperatures. As shown in figure 4a–d, in this framework, a crystal nucleus has been produced when the overall temperature of the system enters the 0.42 $T_m$ isothermal process. Then, in figure 4b–d, two new critical nuclear forms appeared very quickly in succession (about 5 ps apart). Finally, the original generated crystal nuclei grow freely close to each other at 15 ps. At higher temperatures, such as 0.48 $T_m$ in figure 4e–h, the nucleation process is about 5 ps later than 0.42 $T_m$, and a crystal nucleus appears at 5 ps isothermally in figure 4e. Then every 5 ps, new crystal nuclei are generated, and the growth process lags behind 0.42 $T_m$.

Figure 5a adopts the volume density of the number of crystal grains to indicate the number of crystal nuclei at different isothermal times. Only the grains with atomic number greater than 100 are counted in

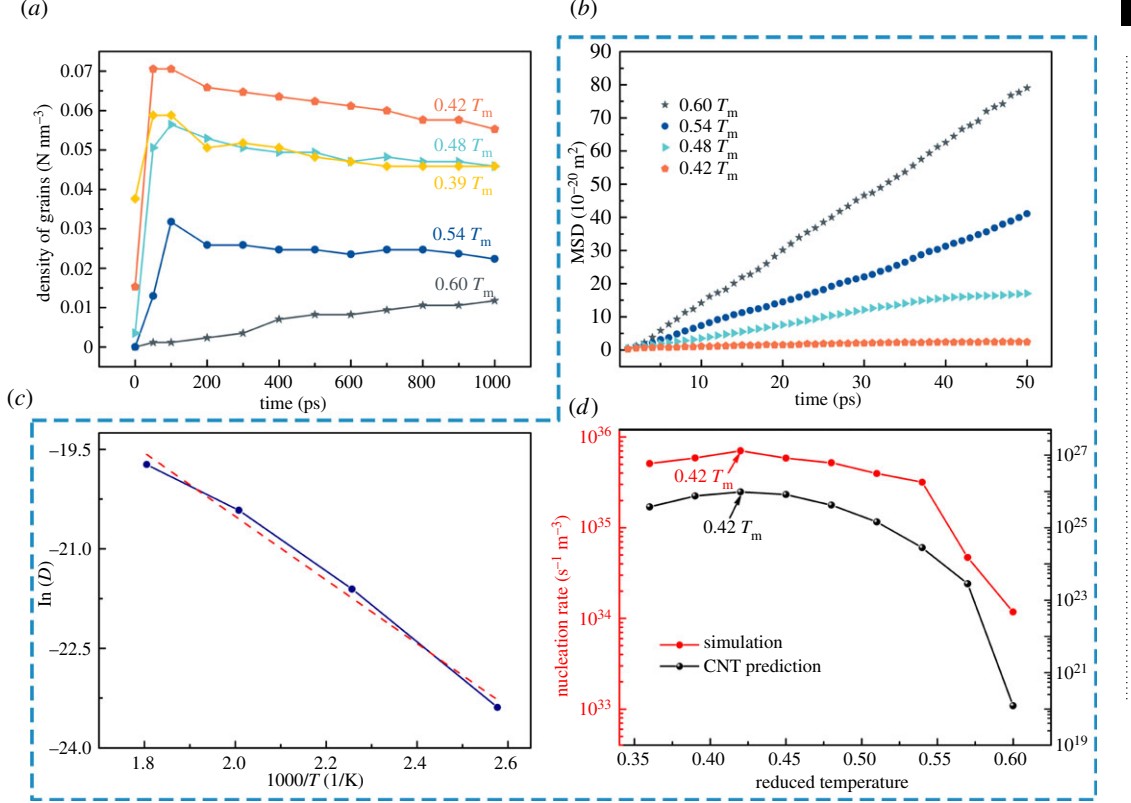

**Figure 5.** The number and volume density of grains and the nucleation rate of solidified structure at different isothermal temperatures. (*a*) The number density of crystal grains changes with time. Except for 0.6 $T_m$, the grain number density increased rapidly in a short period of time and then stabilized. Nucleation mainly occurs at 0–100 ps. (*b*) The change of MSD with time at different temperatures. (*c*) The change of atomic diffusion coefficient with temperature. (*b*) and (*c*) are used to calculate $Q$ in the classical nucleation rate. (*d*) At different isothermal temperatures, Al–4 at.%Cu simulated nucleation rate and CNT predicted nucleation rate change with temperature. Both the simulated and predicted maximum nucleation rates are 0.42 $T_m$.

the grain count statistics. From the change trend of the five curves in figure 5*a*, it can be demonstrated that, except for 0.6 $T_m$, the isothermal nucleation at the other four temperatures is completed before 100 ps, and then individual crystal nuclei are absorbed during the grain growth process. In addition, with the decrease of temperature, the number of nucleation first increased and then decreased, and the number of nucleation is the largest at 0.42 $T_m$.

Acting in accordance with the CNT formula (3.1) [36], The critical nucleation energy $\Delta G^*$ and thermal diffusion activation energy $Q$ are the key factors affecting the nucleation rate $I$.

$$I = I_0 \exp\left(-\frac{\Delta G^*}{kT}\right) \exp\left(-\frac{Q}{kT}\right), \qquad (3.1)$$

where $I_0$ is a constant ($I_0 = n_s^* \varepsilon v_L n$) [36], $n_s^*$ stands for the number of atoms in contact with the liquid on the surface of the critical crystal nucleus (about 100) [36], $\varepsilon$ is the probability of the atom jumping in a given direction (about 1/6) [36], and $v_L$ the vibration frequency of the atoms in the liquid (about $10^{13}$) [36], $n$ is the number of atoms per unit volume [36]. $T$ stands for the temperature, $k$ stands for the Boltzmann constant, $\Delta G^*$ is defined by [36]

$$\Delta G^* = \frac{16\pi\sigma_{SL}^3 T_m^2}{3(\Delta H_m \Delta T)^2}, \qquad (3.2)$$

$\sigma_{SL}$ stands for the interface free energy (0.24 J m$^{-2}$ [37]), $\Delta H_m$ is the enthalpy of melting (10.17 KJ mol$^{-1}$) [38], $T_m$ stands for melting point (923 K) and $\Delta T$ is the undercooling ($\Delta T = T_m - T$).

The thermal diffusion activation energy $Q$ can be obtained by using the mean square displacement (MSD) value calculated by the homogeneous nucleation MD simulation. Figure 5*b* shows the linear relationship of the MSD value changes with time at different temperatures, and the slope increases with temperature. According to the slope of each line, the diffusion coefficient of atoms at different

temperatures can be obtained. Figure 5$c$ is the linear relationship between the diffusion coefficient of the atom and the reciprocal temperature, the red dashed line is a linear fitting line with a slope of $Q/k$ (according to $\ln D = \ln D_0 - Q/(kT)$ [36]). Substituting the calculated values of $\Delta G^*$ and $Q/k$ into formula (3.1) can obtain the nucleation rate $I$.

Figure 5$d$ indicates the variation trend of the nucleation rate with temperature. The red parabola in figure 5$d$ corresponds to the nucleation rate obtained by Al–Cu alloy simulation in this paper. The black parabola in figure 5$d$ corresponds to the predicted value of the CNT. The CNT predicted nucleation rate and the simulated nucleation rate have the same trend with temperature. Following the two curves, the nucleation rate is the lowest at a high temperature of 0.6 $T_m$, and the nucleation rate increases greatly as the degree of subcooling increases. Both curves reach a peak at 0.42 $T_m$, the corresponding simulated maximum nucleation rate is $7.05 \times 10^{35}$ s$^{-1}$ m$^{-3}$, and the predicted CNT nucleation rate is $9.7 \times 10^{25}$ s$^{-1}$ m$^{-3}$. It has been reported that due to the inaccuracy of the interface energy $\sigma_{SL}$ and the constant $I_0$, the CNT predicted nucleation rate differs from the simulated nucleation rate by many orders of magnitude [39]. The same difference of nucleation rate between the predicted value and the simulated value appears in other simulations [28]. But the curve can still show that there must be an optimal critical nucleation temperature near 0.42 $T_m$, at which the number of crystal grains reaches the maximum at the same time. The degree of supercooling continues to increase, and the nucleation rate decreases. The critical nucleation temperature obtained in this paper is also reported in the simulation of homogeneous nucleation of other materials. For instance, the critical temperatures of homogeneous nucleation of Fe and Si are 0.58 $T_m$ [15] and 0.65 $T_m$ [28].

## 3.3. Critical nucleation size

It is known that the classical critical nucleation radius can be obtained by formula (3.3), $\sigma_{SL}$ represents the solid–liquid interface energy, $L_m$ represents the energy absorbed by a unit volume of a substance from the liquid phase to the solid phase, and $\Delta T = T_m - T$. The $\sigma_{SL}$ and $L_m$ of the Al–4 at.%Cu alloy given by Steinbach [37] are 0.24 J m$^{-2}$ and $0.923 \times 10^9$ J m$^{-3}$, respectively. So according to CNT the calculated critical radius (size/diameter) lies between 0.85(1.7) nm and 1.3(2.6) nm for different isothermal temperatures. In the simulation data, the critical crystal nucleus is considered to reach its critical size based on the fact that no atoms in the crystal nucleus return to the liquid. In order to ensure the stability of the simulation data, the isothermal solidification simulation at each temperature is repeated five times. In each simulation, five different crystal nuclei are selected, the diameter length through the centroid of the crystal nucleus is measured every 2°, and the average diameter of the crystal nucleus is calculated. Then define the average value and error bars of the critical core diameter in each case (5-nuclei × 5-replicate simulations). Therefore, the critical nucleation diameter obtained by measurement of MD simulated isothermal nucleation is between 1.75 and 2.92 nm. The relationship between the prediction of the classical nucleation diameter and the simulation result of the critical nucleation diameter and temperature is illustrated in figure 7.

$$r* = \frac{2\sigma_{SL}T_m}{L_m\Delta T}. \tag{3.3}$$

It is discovered from figure 6 that the critical size predicted by CNT is almost similar to the MD simulation at 0.39 $T_m$, 0.42 $T_m$ and 0.54 $T_m$. The MD simulation results come closer to the CNT prediction at lower temperatures (0.39 $T_m$ and 0.42 $T_m$) than at higher temperatures. Since the MD simulation is a quasi-two-dimensional solidification system, the nucleation diameter measured by the plane is larger than the CNT predicted nucleus. However, multiple critical nuclei are formed simultaneously under larger subcooling temperature (for example, solidification at 0.39 $T_m$ and 0.42 $T_m$). In this way, part of the crystal nucleus expansion process is constrained by adjacent nuclei. Therefore, the nucleation at low temperature is similar to the predicted value of classical nucleation.

## 3.4. Crystal nucleus growth and grain coarsening

Figure 7 summarizes two examples of nucleus growth during 0.42 $T_m$ solidification. DXA is employed to colour atoms, green represents FCC structure, red represents HCP structure, white represents amorphous structure (i.e. melt or amorphous). Figure 7$a$ reveals that large crystal nucleus swallows small crystal nucleus of heterogeneous nucleation. Numerous scattered FCC and HCP crystal atoms indicated by blue arrows appeared near the existing crystal nucleus at 8 ps. As time goes by, the number of crystal

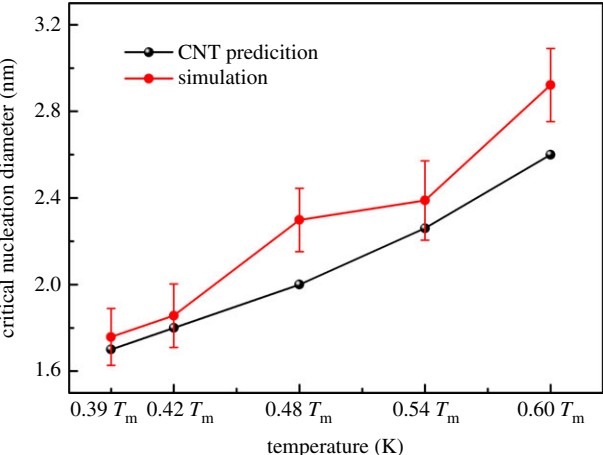

**Figure 6.** The critical nucleus diameter calculated at different temperatures is compared with the results of isothermal nucleation simulation.

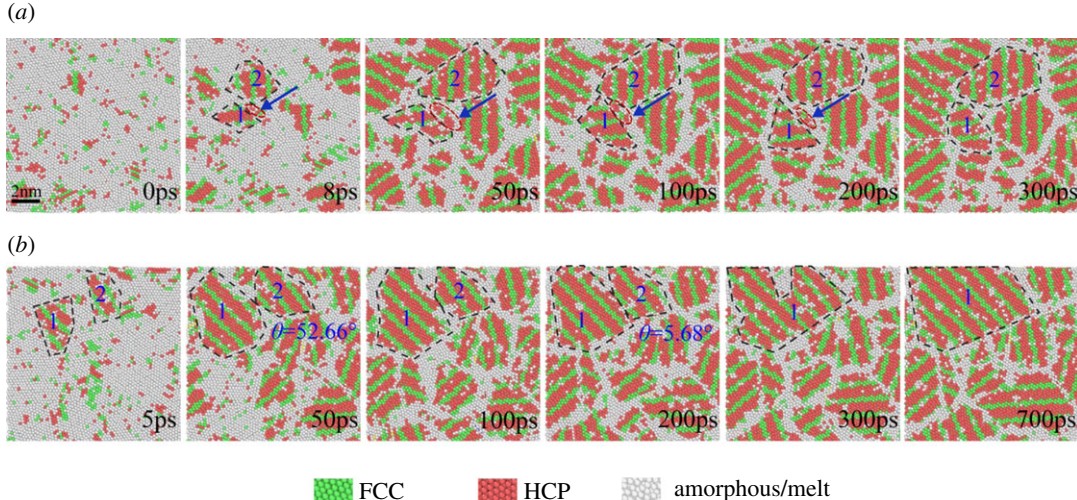

FCC  HCP  amorphous/melt

**Figure 7.** Snapshots of the atomic configuration of the crystal nucleus growth during isothermal solidification at 0.42 $T_m$ in a part of the simulation system. Green represents FCC, red represents HCP, white represents grain boundaries and disordered atoms. (*a*) The small crystal nucleus is attached to the nucleation growth process near the crystal grain; (*b*) two crystal nuclei grow and merge.

atoms continues to increase and gradually aggregates to form a visible crystal nucleus, as shown in the atomic structure diagram of 50 ps. Unfortunately, due to the constraints of the surrounding crystal nucleus, the newly formed crystal nuclei cannot continue to grow, and gradually become smaller after reaching a larger size at 100 ps. Inspecting the size of the small crystal nuclei, it is found that the average diameter of the small crystal nucleus is 0.857 nm, which is smaller than the critical nucleation size at the temperature. Therefore, to 300 ps, the pre-existing crystal grain has completely swallowed the newly generated crystal nucleus. The two homogeneously formed crystal nuclei in figure 7*b* are close to each other at 50 ps. At this time, the misorientation angle $\theta$ is 52.66°. With the elapse of isothermal time, the misorientation angle decreased to 5.68° at 200 ps. At 300 ps, the two crystal nuclei merge into one crystal grain. Then, the grain continues to grow steadily, as captured at 700 ps. According to the DXA analysis of the grain orientation change, it is speculated that this growth mode may be due to the small crystal nucleus being unstable in the initial stage so that the orientation changes during the growth process and gradually approaching the larger grains.

For coarsened crystalline materials, the change of the average diameter of crystal grains with time conforms to the curvature-driven growth theory [40], which can be expressed as

$$\bar{D} \propto k_0 t^n, \tag{3.4}$$

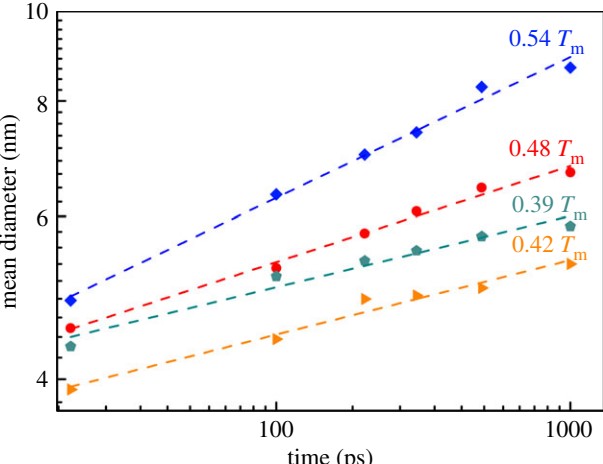

**Figure 8.** The double logarithmic plots of the mean grain diameter as a function of time during the grain growth process at each isothermal temperature.

where $\bar{D}$ is the average diameter of the crystal grains, $t$ is the time, $k_0$ is a constant and n is the grain growth index. The values of $k_0$ and n are determined by the material and temperature, and the crystal growth rate can be represented by a straight line fitted by a logarithmic curve, where n is the slope and $k_0$ is the intercept [41]. In addition, the nucleus growth kinetics also conforms to the curvature-driven growth law [42]. Figure 8 puts forward double logarithmic plots of the average grain diameter at each temperature versus time. Since the number of crystal nuclei under the $0.6\,T_m$ isothermal condition is small, the crystal grain change is not calculated here. According to the slope of the fitted curve of formula (3.4), the maximum grain coarsening index at $0.57\,T_m$ is 0.15, which is less than the theoretical value [40]. However, pure Al has obtained a smaller grain coarsening index (0.06 [43]) during the $0.67\,T_m$ grain coarsening test. Moreover, in the quasi-two-dimensional system simulated by MD, the grain growth index obtained at $0.65\,T_m$ for Si is 0.05 [28], and the grain growth index for Fe at $0.58\,T_m$ is 0.18 [15]. As the temperature decreases in figure 8, the grain coarsening becomes slower and the grain coarsening index gradually decreases, which indicates that temperature is an important factor affecting the grain coarsening. There are many factors that affect the growth of crystals, which will affect the coarsening index of grains, such as grain boundary energy anisotropy [43,44] and the number of grains [40]. In addition, the reason why our grain growth index is small may also be due to the influence of the quasi-two-dimensional system. That is, the mean grain size reaches approximately 10 times the thickness of the calculation system in our calculation, which have affected the result. When reaching 1000 ps, the corresponding crystal nucleus diameters from high temperature to low temperature are 8.389, 6.095, 5.333 and 5.856 nm, respectively. Among them, the crystal grains obtained by isothermal solidification at a temperature of $0.42\,T_m$ are the smallest.

The average atomic potential energy at each isothermal temperature of the simulated system over time is published in figure 9. The potential energy of atoms continues to decrease over time. This is because the grain boundary is related to the excess free energy of activity and is thermodynamically unstable [45]. In the grain growth stage, due to the absorption of small crystal nuclei and the merger of crystal grains, the area of the grain boundary is reduced, and the free energy is reduced accordingly. Therefore, the main driving force for grain growth is the decrease in free energy of GBs [46].

## 3.5. Structure evolution during solidification

The solid phase fraction at each temperature and the percentage of FCC and HCP in the crystal structure obtained from DXA analysis are shown in figure 10. In figure 10a, the solid phase fraction within the same time step increases as the isothermal temperature decreases. At $0.6\,T_m$, the solidification of the Al–Cu alloy is the slowest, and the solid phase fraction is only 60.4% until 1000 ps. With the decrease of isothermal temperature, the solid phase fraction quickly reached the maximum within 250 ps. Isothermal solidification at $0.39\,T_m$, the maximum solid phase fraction within 200 ps is 72.7%. Except for $0.6\,T_m$, when the isothermal temperature of the Al–Cu alloy melt is about 200 ps, the solid phase ratio remains stable, indicating that the melt has all solidified. Figure 10b shows the percentages of FCC and HCP of solidification structure at each temperature, indicating that the solidification

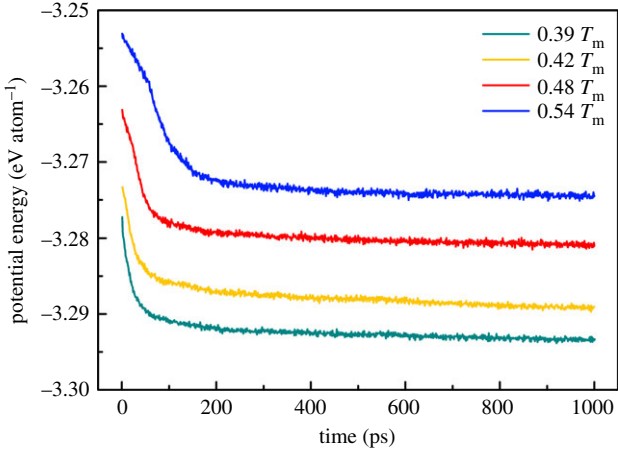

**Figure 9.** The average potential energy of atoms in the simulation system varies with time at each isothermal temperature.

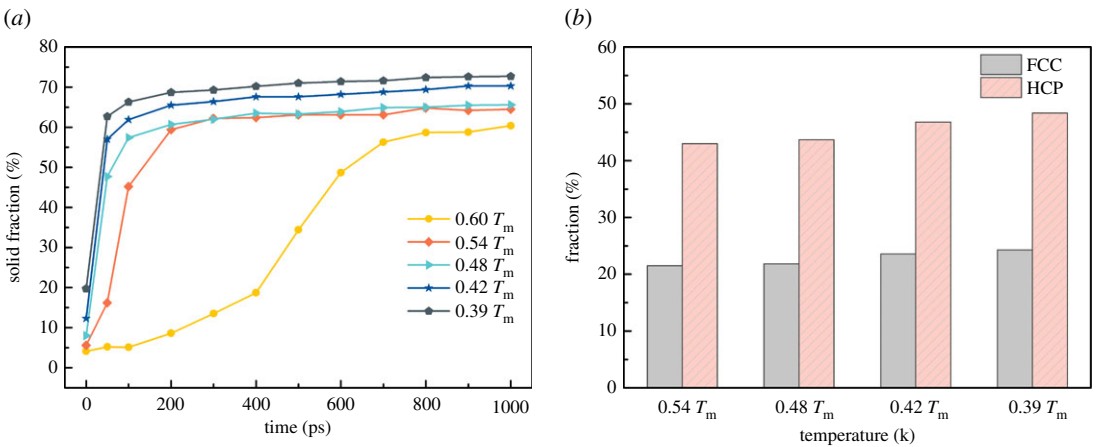

**Figure 10.** The solidification state of the simulation system at different temperatures. (*a*) The solid fraction at different temperatures as a function of time during 1000 ps, and (*b*) the percentage of FCC phase and HCP phase at each isothermal temperature at 1000 ps.

structure of the Al–Cu alloy is composed of FCC and HCP, and HCP is twice that of FCC, and a large number of stacking faults are formed.

Figure 11 reveals the microstructure evolution process of the Al–Cu alloy melt equilibrium solidification at a temperature of 0.42 $T_m$ for 1000 ps, which represents the process of microstructure evolution over time at various temperatures. It can be seen from figure 11 that FCC and HCP grow alternately from the beginning of nucleation, and as the isothermal time continues, the inside of the crystal grains are all layered structures of FCC and HCP. This layered arrangement of FCC and HCP has also appeared in the molecular dynamics simulation of the solidification process of Ni melt [47]. Generally, aluminium-rich aluminium alloys do not have long-period stacking structure due to their high energy state. However, it has recently been reported that when preparing Al/AlN nano-multi-layer films, long-period stacking 9R phases are also found in the Al layer. [48] AlFe alloy film prepared by magnetron sputtering also formed a high-density period stacking phase. [49] And the FCC and HCP in our structure are also arranged regularly, and may also be a long-period stacking structure. Therefore, the specific reasons for the formation of this stacking structure need to be further studied. In addition, the black circles and blue circles in figure 11*d–f* show the area where the grain growth happens and FCC and HCP atoms replaced amorphous solid atoms. As the isothermal solidification continues, the grain boundary movement and amorphous transformation leads to the formation of larger crystal grains. This also explains the reason why the solid phase fraction still slightly increases with the extension of the isothermal time after the crystallization is completed in figure 10*a*. Moreover, at such a low isothermal temperature (0.42 $T_m$), atoms still rely on diffusion motion to reduce GBs, which reduces the total grain boundary energy of the structure. This is consistent with the theory that the grain boundary energy decreases during grain coarsening.

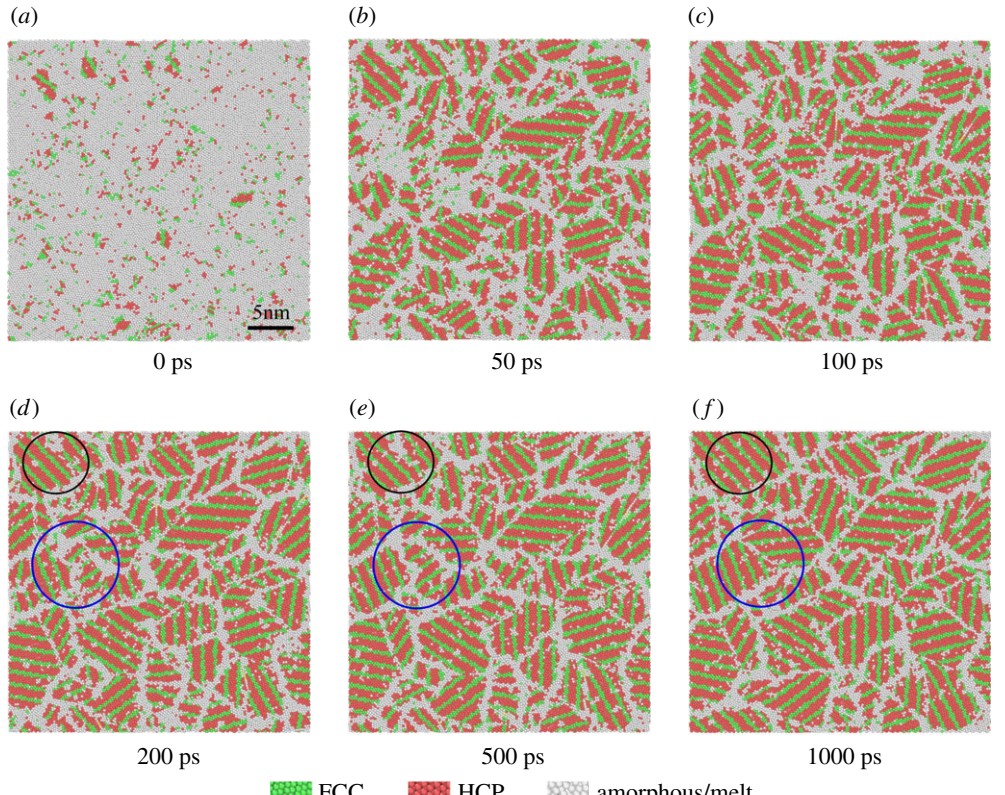

**Figure 11.** Snapshots of the microstructure evolution of Al–4 at.%Cu alloy over time at 0.42 $T_m$. Green represents FCC, red represents HCP, white represents grain boundaries and disordered atoms. (*a*) 0 ps, homogeneous nucleation occurs, and the crystal nucleus is formed by alternate growth of FCC phase and HCP phase. From (*b*) 50 ps to (*c*) 100 ps, the crystal nucleus grows continuously, and the isothermal solidification process is basically completed. From (*d*) 200 ps to (*e*) 500 ps, the grains grow and coarsen. And it reaches stable nanocrystalline structures of FCC and HCP at (*f*) 1000 ps. The black circles and the blue circles in (*d–f*) show the area where the grain growth occurs, and crystal atoms (FCC and HCP) replaced amorphous solid atoms.

## 3.6. Copper atoms distribution

In order to study the distribution of copper atoms during isothermal solidification, we select copper atoms in the PTM snapshot, and the selected atoms are shown in red. We, respectively, show in figure 12 the overall distribution of copper atoms and the distribution in GBs after isothermal 1000 ps at a temperature of 0.48 $T_m$ (443 K). These results suggest that Cu atoms do not produce GB segregation, but occupy Al positions in the crystal structure and are evenly distributed. As mentioned by Zheng *et al.* [50], when the Cu content in the Al matrix is low, the Cu atoms will replace the Al positions in the FCC crystal structure. In order to measure the Cu distribution on the GB, first, all the crystal atoms are removed, and only the GBs are retained, and then we calculated the ratio of Cu atoms to all Cu atoms on the GBs. The result is shown in figure 12*b*. As the isothermal process continues, the atomic percentage of copper on GBs gradually decreases. Combined with figure 12*a*, these results indicate that the growth of crystal grains and the transformation of amorphous atoms to the crystalline state during solidification lower the copper atoms on the GBs, thereby replacing the Al atoms on the FCC or HCP. Basically, when in the solid-state isotherm, the Cu atoms on GBs gradually transform into a crystalline state.

## 4. Conclusion

We performed classical MD simulations employing the EAM interatomic potential. Polycrystalline Al–4 at.% Cu plate-like thin samples are produced by isothermal solidification at different temperatures. In the simulation, multiple isothermal temperatures ranging from 0.6 $T_m$ to 0.39 $T_m$ are adopted, and the isothermal solidification process and microstructure evolution from homogeneous nucleation to crystal

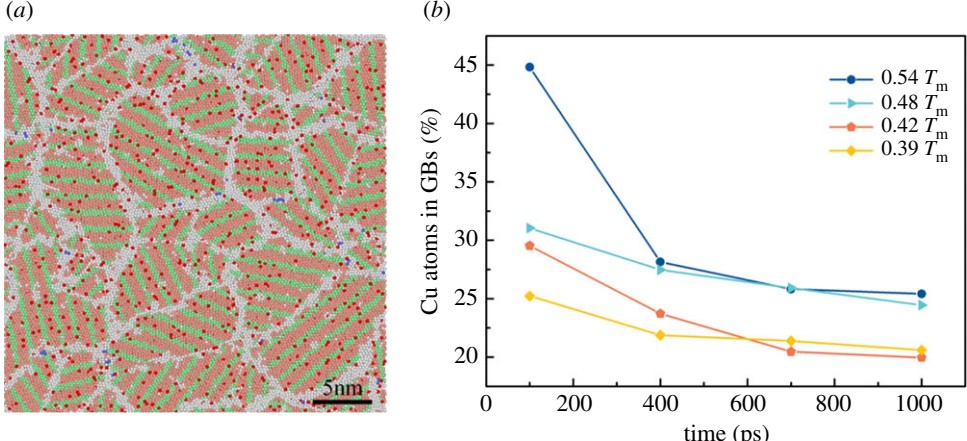

**Figure 12.** At 0.48 $T_m$ (443 K) isothermal 1000 ps (*a*) the overall distribution of copper atoms (red) and (*b*) for each isothermal temperature, the relationship between the percentage of copper atoms in the GBs and the isothermal time.

nucleation growth are investigated. Examining all simulations at different isothermal temperatures, the critical nucleation temperature (about 0.42 $T_m$) of the Al–4 at.%Cu thin-plate sample has been determined, and the nucleation rate is the largest at this temperature. In addition, the average diameter size of the critical nucleus is measured to be about 1.75–2.92 nm during the isothermal process. The size of the critical nucleus follows CNT of prediction value.

During the nucleus growth stage, two growth modes are observed. One is to have large particles swallow tiny heterogeneous nuclei. The other is that when two crystal nuclei grow together, the misorientation of the two crystal grains becomes smaller and smaller over time, and finally merges into one crystal nucleus. And with the progress of grain coarsening, the atomic potential energy is still decreasing, which is consistent with the reduction theory of grain boundary free energy.

For the Al–4 at.%Cu alloy plate-like thin samples created by isothermal solidification, layered nanocrystals composed of alternating FCC and HCP are obtained inside the grains, from the initial stage of nucleation, and this periodic stacking order remains stable until the end of isothermal solidification. Moreover, during the solidification process, the copper atoms gradually replace the Al atoms on the FCC or HCP as the grains grow.

Data accessibility. Data have been uploaded as electronic supplementary material, Excel file.

Authors' contributions. Conceptualization, L.Z. and X.Q.; methodology, L.Z.; software, L.Z.; validation, L.Z. and M.W.; formal analysis, L.Z. and X.Q.; investigation, L.Z. and M.W.; resources, X.Q.; data curation, L.Z.; writing—original draft preparation, L.Z.; writing—review and editing, L.Z. and X.Q.; visualization, L.Z. and M.W.; supervision, X.Q.; project administration, L.Z. and X.Q.; funding acquisition, L.Z. All authors have read and agreed to the published version of the manuscript.

Competing interests. We declare we have no competing interests.

Funding. This study received the support from University Nursing Program for Young Scholars with Creative Talents in Heilongjiang Province of China (grant no. UNPYSCT-2018116).

Acknowledgements. The author thanks the simulation platform provided by the School of Materials Science and Engineering, Jiamusi University.

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
