## [Peer Review File · Royal Society Open Science]

Review History

RSOS-210501.R0 (Original submission)

Review form: Reviewer 1 (Sunil Pradhan)

Is the manuscript scientifically sound in its present form?

Yes

Are the interpretations and conclusions justified by the results?

Yes

Is the language acceptable?

Yes

Do you have any ethical concerns with this paper?

No

Have you any concerns about statistical analyses in this paper?

No

Recommendation?

Accept with minor revision (please list in comments)

Comments to the Author(s)

The manuscript can be accepted for publication in Royal Society Open Science after the authors make the following revisions, arranged in the order of importance.

1. The authors need to discuss more on content of two growth modes for elaborating nucleus growth stage.
2. The terminologies like FCC, HCP need to be written in full form and should be incorporated in the manuscript body.
3. The authors should include more references with respect to isothermal solidification, and periodic stacking.
4. The authors should write melting temperature in Kelvin (K) and not to use T_m in the abstract. If the authors would like to use T_m in the abstract, then T_m meaning needs to be mentioned, so that it will clear to the large readers.
5. The authors need to incorporate more discussion points with respect to Fig.11, for crystal nucleus growth and grain coarsening
- 5.

Review form: Reviewer 2

Is the manuscript scientifically sound in its present form?

Yes

Are the interpretations and conclusions justified by the results?

No

Is the language acceptable?

Yes

Do you have any ethical concerns with this paper?

No

Have you any concerns about statistical analyses in this paper?

No

Recommendation?

Major revision is needed (please make suggestions in comments)

Comments to the Author(s)

This is a paper focusing on the MD simulation of the homogeneous nucleation of growth of crystal in a film structure during solidification. The authors observed the presence of both FCC and HCP phases and analyzed the nucleation rate and growth behavior. They compared the simulation results with the classical nucleation theory and the crystal growth theory.

The following is a list of comments.

1. Can the authors provide the fraction of Cu in the grains/nuclei?
2. The statement "... mainly focused on the analysis of the solidification structure under rapid solidification conditions..." is dubious, since the phase diagram is based on equilibrium state. The study for slow solidification had been done long-long time ago.
3. Please define abbreviations for the first-time use.
4. The authors need to clarify the statement "the research on the homogeneous nucleation mechanism of Al-Cu binary alloy is still inconclusive."
5. The study was focused on the nucleation in a layer structure. The authors need to provide the boundary conditions along the thickness direction. Note that the periodic condition is no good along the thickness direction.
6. According to Fig. 3, there exists an incubation period. The authors please discuss the temperature effect on the incubation period.
7. Please add scale to Figs. 3 and 4.
8. On Fig. 5a, please use the density of grains? Otherwise, the figure is meaningless.
9. On Fig. 5b, please compare the results to the CNT theory.
10. The authors please provide the geometry of the nuclei. Are the nuclei resembled to sphere? If not, Eqs. (2) and (4) need to be modified.
11. Eq. (6) should be $D^n - D_0^n = kt$ with n being 2 or 3. The numerical values of $1/0.05$ is too large. The authors need to check their calculations.

Decision letter (RSOS-210501.R0)

Dear Dr Zhan:

Title: Research on homogeneous nucleation and microstructure evolution of aluminum alloy melt
Manuscript ID: RSOS-210501

The editor assigned to your manuscript has now received comments from reviewers. We would like you to revise your paper in accordance with the referee and Subject Editor suggestions which can be found below (not including confidential reports to the Editor). Please note this decision does not guarantee eventual acceptance.

Please submit your revised paper before 23-May-2021. Please note that the revision deadline will expire at 00.00am on this date. If we do not hear from you within this time then it will be assumed that the paper has been withdrawn. In exceptional circumstances, extensions may be possible if agreed with the Editorial Office in advance. We do not allow multiple rounds of revision so we urge you to make every effort to fully address all of the comments at this stage. If deemed necessary by the Editors, your manuscript will be sent back to one or more of the original reviewers for assessment. If the original reviewers are not available we may invite new reviewers.

To revise your manuscript, log into <http://mc.manuscriptcentral.com/rsos> and enter your Author Centre, where you will find your manuscript title listed under "Manuscripts with Decisions." Under "Actions," click on "Create a Revision." Your manuscript number has been

appended to denote a revision. Revise your manuscript and upload a new version through your Author Centre.

On behalf of the Subject Editor Professor Anthony Stace and the Associate Editor Dr Dattatray Late.

RSC Associate Editor:
Comments to the Author:
Major Revision needed.

RSC Associate Editor:
Comments to the Author:
(There are no comments.)

Reviewers' Comments to Author:
Reviewer: 1

Comments to the Author(s)
The manuscript can be accepted for publication in Royal Society Open Science after the authors make the following revisions, arranged in the order of importance.

1. The authors need to discuss more on content of two growth modes for elaborating nucleus growth stage.
2. The terminologies like FCC, HCP need to be written in full form and should be incorporated in the manuscript body.
3. The authors should include more references with respect to isothermal solidification, and periodic stacking.

4. The authors should write melting temperature in Kelvin (K) and not to use T_m in the abstract. If the authors would like to use T_m in the abstract, then T_m meaning needs to be mentioned, so that it will clear to the large readers.

5. The authors need to incorporate more discussion points with respect to Fig.11, for crystal nucleus growth and grain coarsening

5.

Reviewer: 2

Comments to the Author(s)

This is a paper focusing on the MD simulation of the homogeneous nucleation of growth of crystal in a film structure during solidification. The authors observed the presence of both FCC and HCP phases and analyzed the nucleation rate and growth behavior. They compared the simulation results with the classical nucleation theory and the crystal growth theory.

The following is a list of comments.

1. Can the authors provide the fraction of Cu in the grains/nuclei?
2. The statement "... mainly focused on the analysis of the solidification structure under rapid solidification conditions..." is dubious, since the phase diagram is based on equilibrium state. The study for slow solidification had been done long-long time ago.
3. Please define abbreviations for the first-time use.
4. The authors need to clarify the statement "the research on the homogeneous nucleation mechanism of Al-Cu binary alloy is still inconclusive."
5. The study was focused on the nucleation in a layer structure. The authors need to provide the boundary conditions along the thickness direction. Note that the periodic condition is no good along the thickness direction.
6. According to Fig. 3, there exists an incubation period. The authors please discuss the temperature effect on the incubation period.
7. Please add scale to Figs. 3 and 4.
8. On Fig. 5a, please use the density of grains? Otherwise, the figure is meaningless.
9. On Fig. 5b, please compare the results to the CNT theory.
10. The authors please provide the geometry of the nuclei. Are the nuclei resembled to sphere? If not, Eqs. (2) and (40) need to be modified.
11. Eq. (6) should be $D^n - D_0^n = kt$ with n being 2 or 3. The numerical values of $1/0.05$ is too large. The authors need to check their calculations.

Author's Response to Decision Letter for (RSOS-210501.R0)

See Appendix A.

Decision letter (RSOS-210501.R1)

Dear Dr Zhan:

Title: Research on homogeneous nucleation and microstructure evolution of aluminum alloy melt
Manuscript ID: RSOS-210501.R1

It is a pleasure to accept your manuscript in its current form for publication in Royal Society Open Science. The chemistry content of Royal Society Open Science is published in collaboration with the Royal Society of Chemistry.

On behalf of the Subject Editor Professor Anthony Stace and the Associate Editor Dr Dattatray Late.

RSC Associate Editor
Comments to the Author:
Authors have revised the manuscript as per comments and now suitable for publication.

Reviewer(s)' Comments to Author:

Appendix A

Dear Dr Laura Smith:

Thank you very much for your letter and for the reviewers' comments concerning our manuscript entitled "Research on homogeneous nucleation and microstructure evolution of aluminum alloy melt" (ID: RSOS-210501). Those comments are all valuable and very helpful for revising and improving our paper, as well as the important guiding significance to our researches. We have studied the valuable comments from you and reviewers carefully, and tried our best to revise the manuscript. Revised portion are marked in red in the paper. The main corrections in the paper and the responds to the reviewer's comments are as flowing:

Responds to the reviewer's comments:

Reviewer 1

Thank you very much for your evaluation of our paper and your valuable comments! We have carefully studied your comments and answered your comments. Hope you are satisfied with our responses!

Comment 1: The authors need to discuss more on content of two growth modes for elaborating nucleus growth stage.

Response: We are very grateful to you for your valuable suggestions. We have conducted an in-depth analysis of the nucleus growth methods, and further discussed the reasons for the two growth methods. (For details, see the red part in the first paragraph of section 4.4.)

Comment 2: The terminologies like FCC, HCP need to be written in full form and should be incorporated in the manuscript body.

Response: Thank you for your instructive suggestions. We have adopted the full form in the manuscript body.

Comment 3: The authors should include more references with respect to isothermal solidification, and periodic stacking.

Response: We are very grateful to you for your valuable suggestions. We have added relevant literature in the introduction and discussion. (For example: doi:10.1038/s41467-017-01729-4, doi:10.1016/j.actamat.2019.05.053, doi:10.1063/1.3239469, doi: 10.1134/S0036029518050026, etc.)

Comment 4: The authors should write melting temperature in Kelvin (K) and not to use T_m in the abstract.

If the authors would like to use T_m in the abstract, then T_m meaning needs to be mentioned, so that it will be clear to the large readers.

Response: Thank you for your valuable comment. We have added an explanation of T_m in the abstract.

Comment 5: The authors need to incorporate more discussion points with respect to Fig.11, for crystal nucleus growth and grain coarsening.

Response: We are very grateful to you for your valuable suggestions. We have already conducted an in-depth discussion on Fig.11. For example, in the process of grain growth, amorphous atoms are gradually replaced by FCC and HCP, which increases the degree of crystallization.

Reviewer 2

Thank you very much for your evaluation of our paper and your valuable comments! We have carefully studied your comments and answered your comments. Hope you are satisfied with our responses!

Comment 1: Can the authors provide the fraction of Cu in the grains/nuclei?

Response: Thank you for your instructive suggestion. We have provided in the manuscript the distribution of Cu inside the grains and the percentage of Cu change with time at the grain boundaries during solidification at different isothermal temperatures. (See section 4.6 for details)

Comment 2: The statement "... mainly focused on the analysis of the solidification structure under rapid solidification conditions..." is dubious, since the phase diagram is based on equilibrium state. The study for slow solidification had been done long-long time ago.

Response: Thank you for your careful reading of our manuscript. For this sentence, we may not be accurate enough in expression. We have revised this sentence in the manuscript to "So far, experimental studies on the nucleation and growth behavior of aluminum alloys have mainly adopted optical microscopy (OM), scanning morphology characterization, electron microscopy (SEM) and in-situ transmission electron microscopy to achieve solidification structure analysis."

Comment 3: Please define abbreviations for the first-time use.

Response: We are very grateful to you for your valuable suggestions. We have carefully checked all the abbreviations in the manuscript and defined the abbreviations in the first-time use. Thank you again.

Comment 4: The authors need to clarify the statement “the research on the homogeneous nucleation mechanism of Al-Cu binary alloy is still inconclusive.”

Response: We are very grateful to you for your valuable suggestions. We also think that the expression of this sentence is not accurate enough. Therefore, we have revised this sentence in the manuscript to “Therefore, the study of the solidification law of Al-Cu binary alloy nano-polycrystalline will be of great significance.”

Comment 5: The study was focused on the nucleation in a layer structure. The authors need to provide the boundary conditions along the thickness direction. Note that the periodic condition is no good along the thickness direction.

Response: We are very grateful to you for your valuable suggestions. We very much agree with your point that the layer structure and thin film structure should adopt aperiodic boundary conditions, and its nucleation method should be heterogeneous nucleation. In our research, our system is equivalent to taking a quasi-two-dimensional sample at the center of the actual large-scale solidification structure, which is intended to reveal the details of the homogeneous nucleation process. We have described the system of this study more accurately in the manuscript.

Comment 6: According to Fig.3, there exists an incubation period. The authors please discuss the temperature effect on the incubation period.

Response: Thank you for your instructive suggestion. We have discussed the gestation period of Fig.3 in the manuscript, explaining the relationship between the gestation period and the degree of supercooling. Thank you very much for your valuable comments.

Comment 7: Please add scale to Figs. 3 and 4.

Response: We are very grateful to you for your instructive suggestions. We have added rulers to Fig.3 and Fig.4, and added rulers to other atomic pictures. Thank you very much for your valuable comments.

Comment 8: On Fig.5a, please use the density of grains? Otherwise, the figure is meaningless.

Response: We are very grateful to you for your valuable suggestions. We have revised Fig.5(a) according to your comments.

Comment 9: On Fig. 5b, please compare the results to the CNT theory.

Response: Thank you for your instructive suggestion. We have added the predicted value calculated based on the classical nucleation theory in Fig.5 and compared the predicted value with the simulated value. The simulated nucleation rate has the same trend as the CNT predicted nucleation rate with temperature, and the maximum nucleation rate is obtained at $0.42 T_m$. And we analyzed and discussed the simulated nucleation rate and the predicted nucleation rate in the manuscript.

Comment 10: The authors please provide the geometry of the nuclei. Are the nuclei resembled to sphere? If not, Eqs. (2) and (4) need to be modified.

Response: We are very grateful to you for your valuable suggestions. We have observed that the geometry of the nucleus is similar to a sphere, and detailed information has been added to the manuscript.

Comment 11: Eq. (6) should be $D^n - D_0^n = kt$ with n being 2 or 3. The numerical values of $1/0.05$ is too large. The authors need to check their calculations.

Response: We are very grateful for your valuable suggestions. We have carefully checked the simulation data and the fitting method, and indeed the initial grain size has been neglected in the previous data. We have refitted the average grain data, and the grain coarsening index is 0.15 at $0.57 T_m$. Although the coarsening index is still less than the theoretical value, the grain growth index obtained from the MD quasi-two-dimensional simulation results of polysilicon and iron is also very small, where Si is 0.05 (doi: 10.1039 / C8CE00767E) and Fe is 0.18 (doi: 10.1038 / srep13534). Moreover, it is reported that in the process of grain coarsening of high-purity Al $0.67 T_m$, the grain coarsening index is 0.06 (doi: 10.1103 / PhysRev.71.555). There are many reasons for the small grain growth index obtained by MD simulation. For example, the anisotropy of grain boundary energy and mobility will reduce the grain growth index (doi: org / 10.1016 / 0001-6160 (85) 90093). -8, doi: org/10.1016/S1359-6454(02)00078-2). In addition, the number

of grains will affect the growth index of the grains (doi: org / 10.1016 / 0001-6160 (84) 90151-2). The difference in our situation may also be due to the influence of the quasi-two-dimensional system. In the follow-up work, the physical factors affecting the growth index of the grain index will be studied.

Additional revise:

In addition, as references were added during the revision process, the references were reordered. At the same time, all revisions in the manuscript are marked in red.

We hope that our revised version will be satisfactory for publication in RSOS. Great thanks to you and the referee for the time and effort you expend on this paper.

Thank you.

Come from Dr. Qin.